Metabolites of traffic-related volatile organic compounds in age-related macular degeneration

Mimura Tatsuya mimurat-tky@umin.ac.jp 1 2 3
Noma Hidetaka 4
1 Ophthalmology, Teikyo University School of Medicine , Itabashi-kku , Tokyo , Japan
2 Ophthalmology, Nerima Station West Eye Clinic , Nerima-ku , Tokyo , Japan
3 Ophthalmology, Tsurumi University School of Dentistry , Yokohama , Kanagawa , Japan
4 Ophthalmology, Tokyo Medical University Ibaraki Medical Center , Ami-cho, Inashiki-gun , Ibaraki , Japan
Phairuang Worradorn
Electronic publication date: 2025 Dec 3
Publication date: 2025
Volume: 13
Electronic Location ID: e20405
Received 2025 Jul 30; Accepted 2025 Oct 27
Copyright: ©2025 Mimura et al.
Copyright year: 2025
Copyright holder: Mimura et al.
License: This is an open access article distributed under the terms of the Creative Commons Attribution License, which permits unrestricted use, distribution, reproduction and adaptation in any medium and for any purpose provided that it is properly attributed. For attribution, the original author(s), title, publication source (PeerJ) and either DOI or URL of the article must be cited.
License URL: https://creativecommons.org/licenses/by/4.0/

Keywords: Age-related macular degeneration, Volatile organic compound, 2-methylhippuric acid, 3-methylhippuric acid, Mandelic acid, Phenylglyoxylic acid, Trans trans-muconic acid

Funding: The authors received no funding for this work.

==============================
Background

Volatile organic compounds (VOCs), commonly emitted from vehicle exhaust and industrial activities, are prevalent air pollutants in urban environments. These compounds have been reported to cause various health effects through mechanisms such as oxidative stress and neurotoxicity. Recently, air pollution has attracted attention as a potential risk factor for age-related macular degeneration (AMD); however, the association between VOC exposure and AMD remains unclear.

Objective

This study aimed to assess VOC exposure levels among urban-dwelling AMD patients by quantifying urinary metabolites and investigating the association between VOCs and AMD.

Methods

This cross-sectional study included 40 untreated AMD patients (AMD group), 10 cataract patients (Cataract group), and 10 healthy controls (Healthy group). Representative urinary metabolites of VOCs—2-methylhippuric acid, 3-methylhippuric acid, mandelic acid, phenylglyoxylic acid, and trans,trans-muconic acid—were measured using gas chromatography-mass spectrometry (GC/MS), with concentrations corrected for urinary creatinine. Group comparisons were performed based on creatinine-adjusted metabolite levels.

Results

The AMD group exhibited elevated urinary VOC metabolite levels compared to both control groups. The ratios of mean concentrations in the AMD group versus the Healthy and Cataract groups, respectively, were: 2-methylhippuric acid (201% and 181%), 3-methylhippuric acid (190% and 139%), mandelic acid (304% and 198%), phenylglyoxylic acid (118% and 90%), and trans,trans-muconic acid (214% and 92%). Among these, 2-methylhippuric acid and mandelic acid were significantly higher in the AMD group than in both controls (p < 0.001 and p = 0.042; p < 0.001, respectively). Trans,trans-muconic acid also showed a significant increase compared to the Healthy group (p < 0.001). Correlation analysis within the AMD group revealed moderate but significant associations for 2-methylhippuric acid (r = 0.29, p = 0.011), mandelic acid (r = 0.47, p < 0.001), and trans,trans-muconic acid (r = 0.27, p = 0.020). Multivariate logistic regression identified mandelic acid as an independent factor significantly associated with AMD (odds ratio = 17.20, p < 0.001). Subgroup analysis categorized the AMD group into drusenoid AMD, typical AMD (t-AMD), polypoidal choroidal vasculopathy (PCV), and retinal angiomatous proliferation (RAP). No significant differences in urinary VOC metabolite levels were observed among the four subtypes after multiple comparison adjustment.

Conclusions

These findings indicate that AMD patients are exposed to higher levels of traffic-related VOCs. While mandelic acid—a styrene metabolite—was independently associated with AMD, its role should be interpreted as a potential exposure marker rather than a definitive disease biomarker. Further longitudinal studies are warranted to clarify causal relationships between VOC exposure and AMD development.

Introduction

Age-related macular degeneration (AMD) is a leading cause of irreversible visual impairment among older adults worldwide. Its pathogenesis is multifactorial, involving both genetic and environmental contributors. Environmental exposures, particularly air pollutants and chemical compounds capable of generating reactive oxygen species (ROS), have been increasingly recognized as potential accelerators of these pathological processes. In this context, volatile organic compounds (VOCs)—a major component of urban air pollution—warrant investigation as possible environmental risk factors contributing to AMD onset and progression.

VOCs originating from motor vehicle emissions are among the most ubiquitous environmental pollutants in modern society. Aromatic hydrocarbons such as benzene, toluene, styrene, and xylene are particularly prominent constituents (Montero-Montoya, López-Vargas & Arellano-Aguilar, 2018). These compounds are present in gasoline and exhaust fumes in nanoparticle form, making them readily dispersible into the atmosphere and easily absorbed through inhalation, allowing for systemic distribution within the human body (Qu et al., 2005; Geraldino et al., 2021).

Once absorbed, VOCs undergo metabolic conversion and are primarily excreted in the urine as specific metabolites. For instance, benzene is metabolized to trans,trans-muconic acid, toluene to 2-methylhippuric acid and 3-methylhippuric acid, and styrene to mandelic acid and phenylglyoxylic acid. These urinary metabolites are widely recognized as non-invasive biomarkers for assessing individual VOC exposure (Chaiklieng et al., 2019).

VOCs exposure can be characterized as external exposure, referring to environmental or occupational contact with airborne pollutants, and internal exposure, which reflects the amount of these compounds that has actually entered and been processed within the body. Urinary VOC metabolites serve as reliable indicators of such internal exposure, capturing the integrated effect of inhalation and other transient exposure routes. However, their concentrations can be influenced by factors such as hydration status, renal function, and the timing of urine collection, which may contribute to individual variability in metabolite levels. Although creatinine normalization is often used to correct for these variations, unadjusted urinary concentrations still provide valuable insight into recent VOC uptake and internal body burden.

Previous studies have demonstrated that urinary trans,trans-muconic acid levels increase not only following occupational exposure but also under environmental exposure conditions, such as from tobacco smoke, indicating the sensitivity of this biomarker to benzene inhalation (Lovreglio et al., 2011; Bhandari et al., 2023). In the context of styrene exposure, several studies involving occupationally exposed individuals have reported a strong correlation between urinary mandelic acid levels and exposure intensity. Moreover, mandelic acid has been associated with neurotoxic effects, including impairments in color discrimination and reduced contrast sensitivity (Yuasa et al., 1996; Kishi et al., 2001), suggesting its reliability as a biomarker for styrene-induced health effects.

Styrene is a representative VOC widely used in the plastics and synthetic rubber industries, and it primarily enters the human body via inhalation in occupational settings. Yuasa et al. (1996) reported a significant correlation between urinary mandelic acid concentrations and peripheral nerve conduction velocity—particularly distal latency—among styrene-exposed workers, indicating a dose-dependent neurotoxic effect on peripheral nerve function. Furthermore, Kishi et al. (2001) found that higher urinary mandelic acid (MA) levels were significantly associated with abnormal color vision, particularly tritan (blue-yellow) deficiency, suggesting potential neurotoxic effects on the visual system, including the retina and optic nerve. These findings raise the possibility that VOCs such as styrene may adversely affect not only the central and peripheral nervous systems but also visual function, offering important insights into the ocular toxicity of air pollutants.

AMD is a leading cause of visual impairment among older adults. Beyond genetic susceptibility, environmental factors such as oxidative stress and chronic inflammation have been recognized as major contributors to its pathogenesis (Shaw et al., 2016). Ambati & Fowler (2013) emphasized the integral role of immune responses in AMD pathogenesis. They discussed how inflammation, driven by immune cell activation and cytokine release, contributes to retinal damage and disease progression. This perspective aligns with the understanding that environmental factors, including exposure to industrial pollutants like VOCs, may influence immune mechanisms and thereby modulate AMD risk. Oxidative stress, in particular, is thought to induce chronic damage to retinal pigment epithelium (RPE) cells and photoreceptors, thereby accelerating AMD progression (Plafker, O’Mealey & Szweda, 2012).

VOCs are known to promote oxidative stress through the generation of ROS and depletion of cellular antioxidants like glutathione (Potilinski et al., 2021). Beatty et al. (2000) reported that oxidative stress, resulting from the accumulation of ROS in the retinal tissues, plays a pivotal role in the pathogenesis of AMD. This accumulation can lead to cellular damage, inflammation, and eventual retinal degeneration. Their findings underscore the significance of oxidative stress in AMD development and highlight the potential impact of environmental pollutants, such as VOCs, in exacerbating this condition. Recent epidemiological studies have suggested that traffic-related air pollution may be a contributing factor in the development of AMD (Chua et al., 2022; Ju et al., 2022; Wu, Zhang & Xu, 2024; Sela, Levinshtein & Shulman, 2025). Importantly, although demographic variables such as age and sex did not differ significantly across our study groups (Healthy, Cataract, and AMD), we hypothesize that internal VOC metabolism and systemic exposure burdens may vary by ocular status independently of these demographics. Specifically, we posit that patients with AMD may show distinct patterns of urinary VOC metabolites reflecting either heightened internal absorption, reduced detoxification capacity, or cumulative burden not captured by external ambient monitoring data. To our knowledge, previous epidemiological investigations into AMD and air pollution have largely focused on particulate matter and gaseous pollutants, with limited attention to multiple traffic-related VOC metabolites in untreated AMD patients. By simultaneously quantifying several urinary VOC metabolites (2-methylhippuric acid, 3-methylhippuric acid, mandelic acid, phenylglyoxylic acid, and trans,trans-muconic acid) across three clinical groups, this study aims to provide a more integrated exposure profile and to test the assumption that ocular disease status may be associated with differential internal VOC burden beyond externally measured environmental levels. In doing so, we aim to bridge the gap between external environmental VOC concentrations and ocular disease mechanisms, by testing whether internal VOC metabolite signatures differ in AMD independently of demographic confounders.

Despite these findings, no previous research has comprehensively examined multiple traffic-related VOC metabolites in untreated AMD patients to assess concurrent chemical exposures. A clearer understanding of the relationship between internal VOC exposure and AMD could help bridge the gap between environmental pollution and ocular disease mechanisms and inform future longitudinal studies incorporating repeated sampling and creatinine normalization.

The aim of this study was to assess the level of exposure to traffic-related VOCs in individuals with AMD using urinary metabolites as internal biomarkers. Specifically, we measured the concentrations of 2-methylhippuric acid, 3-methylhippuric acid, mandelic acid, phenylglyoxylic acid, and trans,trans-muconic acid in urine samples, to explore the potential association between vehicular VOC exposure and AMD development.

Methods

Study design

This study was conducted as a prospective, non-randomized, cross-sectional investigation with consecutive case enrollment at Teikyo University Hospital and its affiliated institutions. The study adhered to the principles outlined in the Declaration of Helsinki. The study protocol was approved by the Ethics Committee of Teikyo University School of Medicine (Approval No. 18-228, approved on March 7, 2019). Several studies, including the present one, have been registered as clinical trials in the University Medical Information Network Clinical Trials Registry (UMIN-CTR) (Registration No.: UMIN000013684). Patient recruitment and study procedures were conducted between April 2020 and October 2021. Written informed consent was obtained from all participants prior to enrollment. Figure 1 presents a schematic overview of the experimental protocol.

Figure 1 Schematic overview of the experimental protocol.

Participants

This study enrolled 40 patients who were clinically diagnosed with AMD at Teikyo University and its affiliated hospitals. In cases where both eyes met the inclusion criteria, the eye with poorer visual acuity was selected for evaluation.

The diagnosis and classification of AMD and its subtypes were based on the diagnostic criteria established by the Japanese Retina and Vitreous Society Guidelines Committee for Neovascular Age-Related Macular Degeneration (2025). According to these criteria, participants were classified into the following four subtypes: drusenoid AMD (n = 3), typical AMD (tAMD) (n = 15), polypoidal choroidal vasculopathy (PCV) (n = 19), and retinal angiomatous proliferation (RAP) (n = 3).

Inclusion and exclusion criteria

Patients meeting any of the following criteria were excluded from the study:

(1) age ≤ 50 years;

(2) high myopia (spherical equivalent <−5.0 diopters);

(3) best-corrected visual acuity (BCVA) of 20/20 or better (LogMAR 0.0 or less) in the study eye;

(4) presence of other vitreoretinal diseases, including vitreomacular traction syndrome, epiretinal membrane, macular hole, diabetic retinopathy, or retinal vein occlusion;

(5) history of amblyopia;

(6) nuclear cataract exceeding grade 3 based on the Emery–Little classification;

(7) diagnosis of Alzheimer’s disease or other forms of dementia.

Control groups

Two age-matched control groups were recruited for comparison with the AMD group:

• Ten healthy individuals undergoing routine ophthalmic examinations (Healthy group), and

• Ten patients receiving regular follow-up for cataract without macular pathology (Cataract group).

Fundus examinations and optical coherence tomography (OCT) were performed to confirm the absence of macular disease in both control groups. In the Cataract group, patients with nuclear sclerosis greater than grade 3 according to the Emery–Little classification were included.

All participants were permanent residents of Nerima and Itabashi wards, two adjacent urban districts in Tokyo, Japan. These areas were selected as the study sites due to their comparable demographic characteristics, levels of traffic-related air pollution, and absence of major industrial emission sources. Therefore, the urinary VOC metabolite levels observed in this study are considered to primarily reflect chronic environmental exposure associated with urban living conditions rather than occupational or lifestyle factors.

Clinical data collection

For all participants, demographic data including age, sex, systemic medical history, and past ocular history were obtained at the initial visit. The presence of hypertension, coronary artery disease, and type 1 and type 2 diabetes mellitus was assessed from medical records. Hypertension was defined as the current use of antihypertensive medication or a systolic blood pressure ≥140 mmHg and/or a diastolic pressure ≥90 mmHg at the time of examination. Diabetes mellitus was identified based on self-reported history or use of antidiabetic medications.

The study population consisted of the following:

• Healthy group: 10 participants (four females, six males; mean age 73.3 ± 3.1 years)

• Cataract group: 10 participants (four females, six males; mean age 73.4 ± 6.5 years)

• AMD group: 40 participants (16 females, 24 males; mean age 73.9 ± 7.2 years)

Details are summarized in Table 1.

Urinary metabolite analysis

Automotive gasoline contains aromatic VOCs such as benzene, ethylbenzene, toluene, and xylene (Geraldino et al., 2021). In addition, vehicle exhaust emissions include VOCs such as benzene, toluene, xylene, and styrene, which originate from pyrolyzed naphtha—a byproduct of ethylene and propylene manufacturing (Montero-Montoya, López-Vargas & Arellano-Aguilar, 2018). Once inhaled or absorbed into the human body, these VOCs are metabolized into compounds such as 2-methylhippuric acid, 3-methylhippuric acid, mandelic acid, phenylglyoxylic acid, and trans,trans-muconic acid, which are subsequently excreted in urine (Marchese et al., 2004) (see Fig. 2). In this study, these urinary metabolites were measured as biomarkers of internal VOC exposure.

Figure 2 Metabolic pathways of airborne VOCs and urinary excretion of their metabolites.

Aromatic volatile organic compounds (VOCs), such as toluene, xylene, benzene, ethylbenzene, and styrene, are absorbed into the human body and metabolized into compounds including 2-methylhippuric acid, 3-methylhippuric acid, mandelate, phenylglyoxylate, and trans,trans-muconic acid, which are subsequently excreted in the urine.

Urinary VOC metabolite analysis was performed by US BioTek Laboratories (Seattle, WA, USA) using gas chromatography–mass spectrometry (GC/MS) according to their validated protocol (Salmi et al., 2010). Participants were instructed to abstain from food, beverages, medications, and dietary supplements on the morning of sample collection. First-morning urine samples were collected in disposable paper cups following initial voiding. A standardized collection strip (Dip ‘N Dry, US BioTek, Seattle, WA) was then immersed in each sample. The strips were sealed in foil pouches containing desiccant gel, stored at room temperature in a dry state, and subsequently shipped to US BioTek Laboratories for analysis.

Following the laboratory’s protocol, concentrations of 2-methylhippuric acid, 3-methylhippuric acid, mandelic acid, phenylglyoxylic acid, and trans-trans-muconic acid were measured using GC/MS. In addition, urinary creatinine levels were measured to normalize VOC metabolite concentrations, and all metabolite values were expressed as a ratio to creatinine. Because reference ranges for each metabolite vary by sex, the relative percentage increase or decrease from the sex-specific reference value was calculated for each participant.

All data analyses were conducted in a blinded manner. Investigators performing statistical evaluation were masked to participants’ personal identifiers and group allocation (e.g., disease status).

Analysis of gasoline composition and ambient VOC concentrations

To characterize the chemical composition of automobile fuel used in Japan, we reviewed publicly available product documentation from ENEOS Corporation, one of the country’s major petroleum companies. Specifically, we consulted the Safety Data Sheets (SDS) and technical reports available on the company’s official website (https://www.hd.eneos.co.jp/). The analysis included two commonly distributed fuel types—regular gasoline and premium high-octane gasoline. Information was extracted on the relative content of major aromatic hydrocarbons, including benzene, toluene, ethylbenzene, and xylene (collectively referred to as BTEX), which are known to be key components of VOCs. These data served as a chemical reference for understanding the typical VOC profile emitted by vehicle exhaust, one of the primary sources of urban air pollution. For components listed as concentration ranges in the SDS, representative values were used for interpretation, with due consideration of possible variations across commercial products.

In addition, to understand the background levels of ambient VOC exposure in the residential areas of study participants—specifically, the Itabashi and Nerima wards of Tokyo—we accessed air quality data from the Tokyo Metropolitan Government’s Environmental Monitoring Database (https://www.kankyo.metro.tokyo.lg.jp/). Daily VOC concentration measurements were retrieved from continuous monitoring stations in these districts for a three-year period (January 2019 to December 2021). Annual average concentrations were calculated for key VOCs, including BTEX compounds. All measurements were obtained using automated instruments at fixed monitoring sites and were based on officially published, publicly accessible data.

Statistical analysis

The sample size for this study was determined based on previous reports with comparable study designs (Miyama et al., 2015; Mimura, Noma & Mizota, 2019), which demonstrated statistically significant differences between groups of 10 participants each. Accordingly, we recruited a total of 20 control subjects—10 healthy individuals (Healthy group) and 10 cataract patients (Cataract group). Power analysis was conducted under the following assumptions: 10% margin of error, 95% confidence level, statistical power of 0.8, and a standard deviation of 5. The results indicated that a minimum of eight subjects per group would be sufficient to detect a significant difference between two groups. Therefore, the total of 20 control participants enrolled in this study was deemed statistically adequate.

Comparisons of mean values between two groups were performed using the two-tailed unpaired Student’s t-test. For comparisons involving three or more groups, the Kruskal–Wallis one-way analysis of variance by ranks was employed. Categorical variables were analyzed using either the chi-square test of independence or Fisher’s exact test, depending on expected cell frequencies. Correlations between continuous variables were assessed using Pearson’s correlation coefficient.

To identify independent factors associated with the presence of AMD, forward stepwise multiple logistic regression analysis was conducted. Explanatory variables included the urinary concentrations of five VOC metabolites: 2-methylhippuric acid, 3-methylhippuric acid, mandelic acid, phenylglyoxylic acid, and trans,trans-muconic acid.

All statistical analyses were performed using SAS software, version 9.1 (SAS Institute Inc., Cary, NC, USA). A p-value of less than 0.05 was considered statistically significant.

Results

Patient characteristics

The demographic and clinical characteristics of the study population are presented in Table 1. There were no statistically significant differences among the Healthy, Cataract, and AMD groups with respect to age or sex. Similarly, the prevalence of hypertension (30–47%), type 2 diabetes mellitus (15–20%), and hyperlipidemia (15–20%) did not differ significantly among the groups. No participants in any group reported a history of cerebrovascular disease, coronary artery disease, renal disease, or hepatic disease.

Table 1 Baseline characteristics of the study participants.

	Healthy group	Cataract group	AMD group	P	
Number of patients	10	10	40		
Age (mean ± SD)	73.3 ± 3.1	73.4 ± 6.5	73.9 ± 7.2	*NS	
Sex (female/male)	4/6	4/6	16/24	**NS	
Hypertension	3 (30%)	3 (30%)	19 (47%)	**NS	
Type 2 diabetes mellitus	2 (20%)	2 (20%)	6 (15%)	**NS	
Hyperlipidemia	2 (20%)	2 (20%)	6 (15%)	**NS	
Cerebrovascular disease	0 (0%)	0 (0%)	0 (0%)	**NS	
Coronary artery disease	0 (0%)	0 (0%)	0 (0%)	**NS	
Kidney disease	0 (0%)	0 (0%)	0 (0%)	**NS	
Liver disease	0 (0%)	0 (0%)	0 (0%)	**NS	
Smoking history	3 (30%)	3 (30%)	9 (22%)	**NS	
Notes.

SD, standard deviation; NS, not significant.

* P values were calculated using the Kruskal–Wallis one-way analysis of variance by ranks for comparisons among the three groups.

** Frequencies were analyzed using the chi-square test of independence or Fisher’s exact test, as appropriate.

Regarding smoking history, 30% of individuals in both the Healthy and Cataract groups and 22% in the AMD group reported a history of smoking. However, no statistically significant differences were observed among the three groups (p = 0.818, chi-square test). Furthermore, no participants in any group had a history of glaucoma or diabetic retinopathy.

Although we used the Kruskal–Wallis test, an appropriate non-parametric approach for comparing independent groups with small sample sizes, potential confounding factors such as age, sex, occupational exposure, smoking history, diabetes, hypertension, and renal function were not adjusted in the main analysis. However, no significant intergroup differences were observed in these variables, suggesting that their influence on urinary VOC metabolite levels was limited.

The Cataract group was included as an independent control group distinct from AMD, and subjects were enrolled only if no retinal pathology was observed. Although some participants had nuclear sclerosis above grade III, this was not expected to affect systemic VOC metabolism. Because cataract and AMD are caused by different etiological mechanisms, the use of the Cataract group as a comparator remains appropriate. Future studies with larger sample sizes and multivariable adjustment are warranted to validate these findings.

Composition of gasoline

Table 2 summarizes the concentrations (weight %) of major aromatic hydrocarbons contained in commercially available automotive gasoline in Japan. The average concentrations of individual components in regular and high-octane (premium) gasoline were as follows: benzene, 0.65% and 0.66%; ethylbenzene, 1.1% and 1.4%; xylene, 4.7% and 5.7%; and toluene, 9.0% and 23.0%, respectively. Notably, toluene was present at markedly higher levels in high-octane gasoline compared to other aromatic compounds. This likely reflects the intentional addition of toluene to increase octane ratings. These aromatic hydrocarbons are classified as VOCs, and their high volatility suggests a significant potential for atmospheric release, contributing to urban air pollution.

Table 2 Concentrations of chemical compounds in Japanese gasoline.

	Regular gasoline	High-octane gasoline	
Benzene (%)	0.65	0.66	
Ethylbenzene (%)	1.1	1.4	
Xylene (%)	4.7	5.7	
Toluene (%)	9.0	23.0	
Hexane (%)	3.9	1.1	
Trimethylbenzene (%)	4.3	6.2	
Heptane (%)	1.5	–	
Notes.

Data excerpted from official materials published on the ENEOS Corporation website (https://www.hd.eneos.co.jp/); latest update as of February 1, 2024 (accessed on July 23, 2025).

Ambient VOC concentrations

Table 3 shows the annual average concentrations of VOCs measured between January 2019 and December 2021 in two urban districts of Tokyo (Itabashi and Nerima). The yearly mean concentrations of representative aromatic VOCs ranged as follows: benzene, 0.66–0.83 µg/m3; trichloroethylene, 0.56–1.20 µg/m3; and toluene, 4.90–8.20 µg/m3. Although some variation was observed across years, no substantial year-to-year fluctuations were noted. Among the measured VOCs, toluene consistently showed the highest concentrations in both regions, likely due to emissions from vehicular exhaust and industrial activities. While benzene and trichloroethylene were detected at relatively lower levels, their potential health risks from long-term exposure should not be overlooked.

Table 3 Annual average concentrations of volatile organic compounds (VOCs) in two urban areas of Tokyo (2019–2021).

	Itabashi-ku, Tokyo	Nerima-ku, Tokyo	
	Mean concentration (μg/m3)	Mean concentration (μg/m3)	
Year	2019	2020	2021	2019	2020	2021	
Benzene	0.88	0.70	0.83	0.81	0.66	0.78	
Trichloroethylene	1.20	0.63	0.85	0.87	0.56	0.98	
Tetrachloroethylene	0.31	0.19	0.18	0.24	0.17	0.25	
Dichloromethane	1.70	1.20	1.60	1.60	1.30	1.70	
Acrylonitrile	0.11	0.06	0.12	0.09	0.06	0.09	
Vinyl chloride monomer	0.06	0.03	0.03	0.04	0.02	0.03	
Chloroform	0.26	0.20	0.26	0.29	0.21	0.24	
1,2-Dichloroethane	0.14	0.12	0.12	0.12	0.12	0.11	
1,3-Butadiene	0.24	0.09	0.10	0.18	0.07	0.10	
Ethylene oxide	0.09	0.07	0.08	0.08	0.07	0.07	
Methyl chloride	1.30	1.40	1.40	1.30	1.40	1.50	
Toluene	8.20	6.80	6.90	7.20	4.90	5.70	
Notes.

Daily VOC concentration data were collected from the Air Quality Database published by the Tokyo Metropolitan Government Bureau of Environment (https://www.kankyo.metro.tokyo.lg.jp/climate/), focusing on the residential areas of the study participants (Itabashi and Nerima Wards, Tokyo). The average annual concentrations were calculated for each year from January 2019 to December 2021. The data presented represent annual arithmetic means, and the retrieval date was July 22, 2025.

There were no significant differences in VOC concentrations between the two districts, suggesting a shared background of urban air pollution across the study population’s residential areas. These environmental measurements provide important contextual data supporting the relevance of VOC exposure in the lives of the enrolled patients and serve as a foundational basis for interpreting the clinical associations observed in this study.

Urinary VOC metabolite concentrations and percent increases relative to reference means across study groups

Figure 3 displays mean urinary concentrations (±SD) of VOC metabolites, normalized to creatinine, alongside their percent changes relative to established reference mean values in the Healthy, Cataract, and AMD groups.

Figure 3 Comparison of urinary concentrations of VOC metabolites (left panels: mean ± standard deviation) and their relative increase compared to reference values (right panels: box plots) among the Healthy (n = 10), Cataract (n = 10), and AMD (n = 40) groups.

(A) 2-Methylhippuric acid; (B) 3-Methylhippuric acid; (C) Mandelate; (D) Phenylglyoxylate; (E) trans,trans-Muconic acid. Two-group comparisons were conducted using the two-tailed unpaired Student’s t-test, and comparisons among three groups were performed using the Kruskal–Wallis test. Individual data points were not displayed to maintain figure clarity.

• 2-Methylhippuric acid (Fig. 3A):

∘ Healthy: 2.98 ± 0.83 µg/mg creatinine, Δ −0.1 ± 12.4%

∘ Cataract: 3.31 ± 3.76 µg/mg, Δ 5.1 ± 63.7%

∘ AMD: 5.99 ± 4.67 µg/mg, Δ 69.9 ± 75.7% The AMD group exhibited 201% and 181% higher mean concentrations compared to the Healthy and Cataract groups, respectively. Group differences were statistically significant (p = 0.035, Kruskal–Wallis). Post-hoc analyses confirmed significantly elevated levels in AMD versus Healthy (p < 0.001) and versus Cataract (p = 0.042) using two-tailed unpaired t-tests.

• 3-Methylhippuric acid (Fig. 3B):

∘ Healthy: 0.17 ± 0.12 µg/mg, Δ 18.6 ± 52.0%

∘ Cataract: 0.23 ± 0.22 µg/mg, Δ 36.9 ± 90.9%

∘ AMD: 0.31 ± 0.68 µg/mg, Δ 84.1 ± 294.8% The AMD group showed 190% and 139% higher means relative to the Healthy and Cataract groups. However, differences across the three groups were not significant (p = 0.395, Kruskal–Wallis), nor were pairwise differences (AMD vs. Healthy, p = 0.105; AMD vs. Cataract, p = 0.254).

• Mandelic acid (Fig. 3C):

∘ Healthy: 0.08 ± 0.06 µg/mg, Δ –22.4 ± 25.4%

∘ Cataract: 0.12 ± 0.07 µg/mg, Δ −3.9 ± 29.8%

∘ AMD: 0.24 ± 0.14 µg/mg, Δ 41.8 ± 57.6% The AMD group had concentrations 304% and 198% higher than Healthy and Cataract groups. Overall differences were significant (p = 0.035, Kruskal–Wallis), with AMD significantly higher than both comparison groups (p < 0.001).

• Phenylglyoxylic acid (Fig. 3D):

∘ Healthy: 0.19 ± 0.07 µg/mg, Δ 0.5 ± 17.8%

∘ Cataract: 0.25 ± 0.13 µg/mg, Δ 17.1 ± 34.9%

∘ AMD: 0.23 ± 0.11 µg/mg, Δ 9.5 ± 30.0% Although mean ratios were 118% (vs. Healthy) and 90% (vs. Cataract), there were no significant differences among groups (p = 0.607, Kruskal–Wallis).

• trans, trans-muconic acid (Fig. 3E):

∘ Healthy: 0.02 ± 0.02 µg/mg, Δ –10.1 ± 12.5%

∘ Cataract: 0.06 ± 0.04 µg/mg, Δ −0.1 ± 28.9%

∘ AMD: 0.05 ± 0.03 µg/mg, Δ –10.2 ± 22.9% Despite a 214% higher mean in AMD relative to Healthy, and 92% relative to Cataract, the difference across groups was significant (p = 0.025, Kruskal–Wallis). AMD was significantly higher than Healthy (p < 0.001) but not Cataract (p = 0.379).

Comparison of urinary VOC levels among AMD subtypes

Figure 4 presents urinary metabolite levels and their percent increases relative to reference means across AMD subtypes: drusenoid, typical (t-AMD), PCV, and RAP. Metabolites evaluated included 2-methylhippuric acid (Fig. 4A), 3-methylhippuric acid (Fig. 4B), Mandelic acid (Fig. 4C), Phenylglyoxylic acid (Fig. 4D), and trans,trans-muconic acid (Fig. 4E). No significant differences were found among subtypes for any metabolite (Kruskal–Wallis).

Figure 4 Comparison of urinary concentrations and relative increase ratios of VOC metabolites among the four AMD subtypes.

The AMD group was subdivided into four clinical subtypes: drusenoid, typical AMD (t-AMD), polypoidal choroidal vasculopathy (PCV), and retinal angiomatous proliferation (RAP). Urinary concentrations (mean ± standard deviation) and relative increase compared to reference values were compared for the following metabolites: (A) 2-Methylhippuric acid, (B) 3-Methylhippuric acid, (C) Mandelate, (D) Phenylglyoxylate, and (E) trans,trans-Muconic acid. Statistical analysis among the four groups was conducted using the Kruskal–Wallis test.

The proportion of AMD cases exceeding reference values (i.e., %Δ >0) for each metabolite was:

• 2-methylhippuric acid: 90.0% (36/40)

• 3-methylhippuric acid: 37.5% (15/40)

• Mandelic acid: 77.5% (31/40)

• Phenylglyoxylic acid: 57.5% (23/40)

• trans,trans-muconic acid: 32.5% (13/40)

Correlation among urinary VOC metabolites

Figure 5A displays pairwise Pearson correlation coefficients among the five derivates, and Fig. 5B visualizes them via a bubble plot. Significant positive correlations were identified in six metabolite pairs:

Figure 5 Correlation analysis of urinary concentrations of VOC metabolites.

(A) Pairwise correlations of urinary concentrations among five VOC metabolites: 2-methylhippuric acid, 3-methylhippuric acid, mandelate, phenylglyoxylate, and trans,trans-muconic acid. (B) Three-dimensional bubble plot visualizing the Pearson correlation coefficients between each pair of metabolites.

• 2-methylhippuric acid & mandelic acid (r = 0.44, p < 0.001)

• 2-methylhippuric acid & phenylglyoxylic acid (r = 0.32, p = 0.012)

• 3-methylhippuric acid & trans,trans-muconic acid (r = 0.29, p = 0.027)

• Mandelic acid & phenylglyoxylic acid (r = 0.56, p < 0.001)

• Mandelic acid & trans,trans-muconic acid (r = 0.38, p = 0.003)

• Phenylglyoxylic acid & trans,trans-muconic acid (r = 0.32, p = 0.012)

No significant correlations were observed between:

• 2-methylhippuric acid & 3-methylhippuric acid (r = 0.05, p = 0.681)

• 2-methylhippuric acid & trans,trans-muconic acid (r = 0.11, p = 0.387)

• 3-methylhippuric acid & mandelic acid (r = 0.11, p = 0.412)

• 3-methylhippuric acid & phenylglyoxylic acid (r = 0.07, p = 0.581)

These findings may reflect shared metabolic pathways or common exposure sources among certain VOCs.

Association of VOC metabolites with AMD

Pearson correlation and forward stepwise logistic regression analyses were used to assess associations between urinary VOC metabolites and AMD status. Results (Table 4) revealed significant positive correlations in the AMD group for:

Table 4 Correlation and multivariate analysis of urinary VOC metabolites in relation to AMD (n = 60).

	Correlation coefficient	Multivariate analysis	
Variable	R	(95% CI)	P Value	OR	(95% CI)	P Value	
1. 2-Methylmal uric acid	0.29	(0.04–0.51)	0.011	1.03	(−0.02–0.07)	0.261	
2. 3-Methylmal uric acid	0.10	(−0.16–0.35)	0.218	1.03	(−0.29–0.34)	0.874	
3. Mandelate	0.47	(0.25–0.65)	<0.001	17.20	(1.21–4.48)	<0.001	
4. Phenylglyoxylate	0.08	(−0.18–0.32)	0.279	0.12	(−4.05–0.24)	0.028	
5. Trans-trans-Muconic acid	0.27	(0.01–0.49)	0.020	30.59	(−2.66–9.50)	0.264	
6. Mandelate+ Phenylglyoxylate (4+5)	0.14	(−0.12–0.38)	0.148	–	–	–	
7. All (1+2+3+4+5)	0.31	(0.07–0.53)	0.007	–	–	–	
Notes.

To evaluate the association between urinary VOC metabolite levels and the presence of AMD, both correlation analysis and multivariate regression analysis were performed. Pearson’s correlation coefficients were calculated using a two-tailed Pearson’s product-moment correlation formula to assess linear relationships between VOC concentrations and AMD-related group differences. In addition, stepwise multivariate logistic regression analysis (forward selection method) was conducted to identify independent VOC metabolites significantly associated with AMD prevalence. R, Pearson’s correlation coefficient; CI, confidence interval; OR, odds ratio.

• 2-methylhippuric acid (r = 0.29, p = 0.011)

• Mandelic acid (r = 0.47, p < 0.001)

• trans,trans-muconic acid (r = 0.27, p = 0.020)

No significant correlations were found for 3-methylhippuric acid (r = 0.10, p = 0.218) or Phenylglyoxylic acid (r = 0.08, p = 0.279).

The cumulative measure for all five VOCs also showed a significant correlation with AMD status (r = 0.31, p = 0.007).

In multivariate logistic regression, mandelic acid remained a significant positive predictor of AMD (odds ratio (OR) = 17.20, p < 0.001), whereas phenylglyoxylic acid was identified as a significant negative predictor (OR = 0.12, p = 0.028).

Discussion

Summary of the study

This study aimed to investigate the association between VOC metabolites and AMD by comparing urinary concentrations of VOC metabolites among three groups: Healthy, Cataract, and AMD. The results revealed that urinary levels and percent increases relative to reference values for 2-methylhippuric acid, mandelic acid, and trans,trans-muconic acid were significantly elevated in the AMD group. These findings suggest a possible involvement of these compounds in the pathogenesis of AMD.

In particular, both 2-methylhippuric acid and mandelic acid were significantly higher in the AMD group compared to the Healthy and Cataract groups. They also showed significant positive correlations with AMD in Pearson’s analysis. Notably, mandelic acid emerged as a significant independent factor associated with AMD in multivariate regression analysis, suggesting its potential utility as a biomarker for AMD risk. Conversely, although phenylglyoxylic acid tended to be higher in the AMD group, the differences were not statistically significant, indicating a limited association with AMD. Several urinary VOC metabolites did not show statistically significant differences between groups, but these findings are still valuable in understanding the overall exposure profile. The multivariate analysis indicated that only mandelic acid was independently associated with AMD after adjustment for potential confounders, suggesting that styrene-related exposure may play a unique role in AMD pathogenesis. The coexistence of both significant and non-significant results reflects the complex multifactorial nature of AMD, in which environmental and metabolic influences may vary among individuals.

The gasoline commonly used in the residential areas of study participants was found to contain high concentrations of toluene, xylene, ethylbenzene, and benzene. Air quality monitoring data from two urban areas in Tokyo further confirmed the presence of VOCs in the environment, with toluene detected at the highest levels, followed by benzene and trichloroethylene. These compounds are known to be metabolized into the urinary biomarkers analyzed in this study: methylhippuric acids (from toluene and xylene), mandelic acid (from ethylbenzene), and trans,trans-muconic acid (from benzene).

These VOCs are well-documented components of urban air pollution, primarily originating from automobile exhaust and industrial emissions. Chronic inhalation exposure has been shown to induce oxidative stress by generating ROS and depleting intracellular glutathione, a key antioxidant (Wang et al., 2013). The significant elevation of these metabolites in the AMD group observed in this study suggests that long-term VOC exposure may contribute to oxidative damage of RPE cells and photoreceptors, thereby playing a role in the onset and progression of AMD.

Correlation analyses among the metabolites revealed particularly strong positive associations between 2-methylhippuric acid and mandelic acid, mandelic acid and phenylglyoxylic acid, and mandelic acid and trans,trans-muconic acid. These findings imply that certain VOC metabolites may share common metabolic pathways or originate from overlapping environmental exposure sources. In contrast, 3-methylhippuric acid exhibited substantial inter-individual variability and showed no significant differences among groups or clear association with AMD.

Importantly, no significant differences in age, sex, smoking history, or systemic comorbidities such as type 2 diabetes and hypertension were observed among the study groups. Therefore, the elevated urinary VOC metabolite levels in the AMD group are likely to reflect environmental VOC exposure rather than underlying lifestyle or medical conditions.

Proposed mechanisms linking VOC exposure to AMD progression

The elevated urinary levels of 2-methylhippuric acid, mandelic acid, and trans,trans-muconic acid observed in the AMD group suggest increased systemic exposure to aromatic VOCs such as toluene, ethylbenzene, and benzene. These compounds are commonly present in urban air due to traffic and industrial emissions, and chronic exposure to them has been linked to tissue oxidative injury and inflammatory responses in multiple organs. Considering the established metabolic origins of these urinary biomarkers, their elevation provides indirect but biologically meaningful evidence of environmental VOC exposure. This association supports the plausibility that long-term exposure to these airborne compounds contributes to retinal tissue damage through oxidative stress, complement-mediated inflammation, and ocular surface irritation.

Based on the findings of this study, several mechanistic pathways are proposed through which traffic-related VOCs—including benzene, toluene, styrene, and xylene—may contribute to the development and progression of AMD.

First, oxidative stress induced by VOC exposure may play a critical role in damaging RPE cells. VOCs are inhaled and metabolized in the body, during which ROS are generated. It has been reported that compounds such as toluene, benzene, and styrene activate the p38 MAPK signaling pathway, upregulating cyclooxygenase-2 (COX-2) expression and promoting the production of pro-inflammatory prostaglandins such as PGE2 and PGF2α, thereby inducing oxidative stress and inflammation (Mögel et al., 2011). Given the high oxygen demand and rich content of polyunsaturated fatty acids in the RPE, these cells are particularly vulnerable to oxidative damage. Such stress may trigger cellular injury, drusen formation, and the activation of pro-inflammatory signaling cascades (Ochoa Hernández et al., 2025). Although oxidative stress parameters were not directly evaluated in this study, previous reports have established oxidative stress as a key mechanism by which VOC exposure induces cellular and retinal injury (Potilinski et al., 2021; Chua et al., 2022). Therefore, the discussion of oxidative stress in the present study is intended to provide biological context rather than to imply a direct causal inference.

Second, chronic inflammation induced by long-term VOC exposure may contribute to AMD pathogenesis through the activation of inflammatory cytokines and the complement cascade. VOCs not only cause oxidative stress but are also known to modulate immune responses. Several experimental studies have shown that VOCs upregulate COX-2 and inflammatory cytokines in macrophages and epithelial cells, particularly in pulmonary and other mucosal tissues, thus acting as a trigger for chronic inflammation (Mögel et al., 2011). In AMD, chronic inflammation—mediated by complement activation and microglial recruitment—is recognized as a central pathogenic feature. Genetic polymorphisms, such as those in complement factor H (CFH), may exacerbate inflammation and accelerate disease progression (Ogbodo et al., 2022).

Third, VOCs may contribute to retinal damage indirectly through effects on the ocular surface. VOCs present in the environment can adhere to the cornea and conjunctiva, inducing local irritation, dryness, inflammation, and oxidative stress. Indeed, symptoms such as eye dryness and irritation, along with increased oxidative stress markers (e.g., elevated O-methylhippuric acid), have been reported in individuals with “sick building syndrome” linked to indoor VOC exposure (Kwon et al., 2018). Persistent exposure at the ocular surface may allow inflammatory or oxidative signals to propagate inward, potentially affecting deeper ocular tissues such as the retina.

Previous clinical trials, such as the Age-Related Eye Disease Study 2 (AREDS2), demonstrated that antioxidant and nutraceutical supplementation—particularly with lutein, zeaxanthin, and omega-3 fatty acids—can attenuate oxidative tissue damage and slow AMD progression (Chew et al., 2014). These findings suggest that modulation of oxidative stress through dietary or pharmacological measures could potentially mitigate retinal damage resulting from chronic VOC exposure.

Taken together, these findings suggest that VOCs may contribute to AMD progression through multiple interconnected mechanisms: (1) oxidative stress-mediated damage to RPE cells, (2) chronic inflammation via activation of inflammatory cytokines and the complement system, and (3) ocular surface irritation potentially influencing retinal health. In essence, VOC-induced oxidative stress and chronic inflammation may play a critical role in RPE and photoreceptor degeneration, contributing to the pathophysiology of AMD.

Future research should aim to identify specific VOC exposure sources, establish dose–response relationships, and integrate oxidative stress biomarkers with retinal imaging findings to clarify the causal link between VOC exposure and AMD development.

Limitations

This study has many limitations

First, confounders such as occupational and domestic exposure factors, proximity to heavily trafficked roadways, inadequate household ventilation, and passive tobacco smoke inhalation cannot be completely ruled out. Although participants with a history of active smoking were balanced among groups, residual confounding by environmental or lifestyle factors may still have influenced the results. Future studies should include these variables in the study design and conduct sensitivity analyses excluding active smokers, while appropriately adjusting for age, sex, and systemic comorbidities to minimize heterogeneity and strengthen causal inference.

Second, there is currently no established method to directly measure VOC concentrations within the intraocular environment.

Third, VOC exposure was assessed indirectly through urinary metabolites, which limits our ability to precisely quantify exposure routes, dosage, and duration.

Fourth, due to the cross-sectional nature of this study, a causal relationship between VOC exposure and AMD development cannot be definitively established.

Fifth, while the AMD group was subclassified, no significant differences in VOC metabolite levels were observed among subtypes. This suggests that a larger sample size may be required to detect subtype-specific effects.

Another limitation of the present study is that the urinary analyses of VOC metabolites were conducted by a commercial laboratory (US BioTek Laboratories, Shoreline, WA, USA), and detailed analytical characteristics such as internal standards, calibration curves, limits of detection, and coefficients of variation were not available to the authors. In addition, the study did not evaluate potential diurnal variations in metabolite excretion or the risk of exposure misclassification due to differences in exposure routes. These factors may have introduced variability and should be considered when interpreting the findings.

Conclusion

This study demonstrated that urinary levels of specific VOC metabolites were significantly elevated in patients with AMD, suggesting a possible association between urban VOC exposure and AMD onset.

In particular, concentrations of mandelate and 2-methylhippuric acid were significantly higher in the AMD group compared to other groups, with mandelate emerging as an independent factor associated with AMD. These findings point to its potential utility as a biomarker of environmental exposure.

Our results further support the hypothesis that chronic VOC exposure may contribute to AMD progression through multiple mechanisms, including oxidative stress, sustained inflammation, and possible indirect effects via the ocular surface.

Modern insights into AMD pathogenesis have underscored the central role of inflammation and complement activation in retinal degeneration (Ambati & Fowler, 2013). In this context, environmental oxidative stressors such as VOCs may serve as external triggers that exacerbate complement-mediated inflammation, thereby linking exposure risk with established pathogenic pathways in AMD.

Future studies are warranted to more precisely evaluate exposure pathways and levels, and to explore how VOCs may differentially impact AMD subtypes. To promote transparency and reproducibility, the anonymized dataset containing individual urinary metabolite measurements will be made openly available following publication. This open-access data provision will allow independent validation and meta-analytic integration with future studies, contributing to standardized methodologies and more reproducible outcomes in environmental ophthalmology research. Future longitudinal studies are planned to combine timed urinary metabolite assessments with quantitative retinal imaging and free radical measurements. Such integrative approaches will enable detailed analysis of exposure–response relationships and improve causal inference by linking biochemical markers of oxidative stress to structural retinal alterations. Ultimately, randomized controlled trials incorporating both biological and imaging endpoints will be required to determine whether reducing VOC exposure can alter the clinical course of AMD. Nonetheless, our findings provide novel insights into environmental risk factors for AMD and may inform new strategies for disease prevention in populations exposed to urban air pollution.

Supplemental Information

Supplemental Information 1 The CONSORT PRO Reporting Guidance Checklist

Supplemental Information 2 STROBE Checklist

Supplemental Information 3 Protocol (Japanese)

Supplemental Information 4 Protocol (English)

Supplemental Information 5 Raw data

Supplemental Information 6 Raw data for Table 1

Supplemental Information 7 Raw data for Table 2

Additional Information and Declarations

Competing Interests

Author Contributions

Human Ethics

Clinical Trial Ethics

Data Availability

Clinical Trial Registration

Tatsuya Mimura is employed by Nerima Station West Eye Clinic. The authors declare no other competing interests.

Tatsuya Mimura conceived and designed the experiments, performed the experiments, analyzed the data, prepared figures and/or tables, authored or reviewed drafts of the article, and approved the final draft.

Hidetaka Noma conceived and designed the experiments, performed the experiments, analyzed the data, prepared figures and/or tables, authored or reviewed drafts of the article, and approved the final draft.

The following information was supplied relating to ethical approvals (i.e., approving body and any reference numbers):

Ethical approval was obtained from the Ethical Review Committee of Teikyo University (Approval No. 18-228).

The following information was supplied relating to ethical approvals (i.e., approving body and any reference numbers):

The Ethical Review Committee of Teikyo University (Registration No.: UMIN000013684).

The following information was supplied regarding data availability:

The raw data are available in the Supplemental Files.

The following information was supplied regarding Clinical Trial registration:

UMIN000013684.

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
