# Peer review of "Metabolites of traffic-related volatile organic compounds in age-related macular degeneration"

_PeerJ, doi:10.7717/peerj.20405_

## Round 0.1 · original submission · Major Revisions

· Academic Editor

Major Revisions

This unique study links evidence-based doses of volatile organic compounds to AMD in the ophthalmic community. A single revision cycle can improve the paper's methods without changing the narrative. Ethical concerns, registry statements, and research design align with the goals. Sensitivity analysis and prospective statements help address the major risks of multivariable modeling, including a small sample size and potential confounders. Major improvements to the visuals and tables should make the manuscript publishable.

Reviewer 1 ·

Basic reporting

Title & Abstract
The research title delineates several keywords to highlight its focus on age-related macular degeneration (AMD) as a disease condition. The title particularly suggests a key research query on the role of volatile organic compound metabolites in AMD. The abstract encapsulates the background, objective, design, and key findings, specifying a single metabolite: mandelic acid with AMD. The detection of higher exposure of patients with AMD to this chemical compound adds to its scientific novelty; thus, group-based comparisons highlighted in the abstract’s concise methodology (on page 8) are ideal for the study.

The authors have to provide some clarification regarding the sample size selection. Some justification is also needed concerning the statistical methods employed to distinguish between the risk of association and the risk of causation. To address these aforementioned concerns, the following sentence can be placed as a third paragraph of the abstract’s Methods section. The study’s sample size was pre-determined with data provided from prior studies with analogous designs, aimed at attaining group similarities near the eighty percentile in creatinine-adjusted metabolite concentrations.

Introduction
The authors investigate an understudied paradigm relating chemical compound exposure to AMD. This study’s clinical relevance is tangible, and it employs a well-structured systematic approach. Aspects in need of minute to broader adjustments include the use of precise biochemical analysis techniques, justifying exclusion criteria for control participants with cataracts, robust statistical methodology via non-parametric tests, and the elucidation of confounders among study groups. The ambient monitoring context is a significant asset and might be utilized by using a straightforward exposure contrast that associates participant postal codes with district-level averages as a descriptive sensitivity analysis. Minor typographical errors must be rectified throughout the captions and text.

The study’s background section provides information about the occurrence of neuro-visual effects of styrene and refers to prior studies linking styrene exposure to AMD. The study’s aim is delineated at the close of this section, with a list of metabolites appended. This section also comprises an assessment of exposure risk and AMD disease-related biomarkers.
To enhance this section, exposure risk should be clarified by contrasting between determinants of extrinsic exposure and intrinsic exposure to organic compounds. The authors can also provide further clarity regarding the correlation between urinary metabolite concentration, as well as time of exposure and routine rehydration, even with the absence of creatinine normalization metrics. Urinary metabolites make up the biomarker of intrinsic exposure, accounting for routes such as inhalation and other forms of temporary toxin exposure. This may also signal person-to-person differences in fluid balance and renal function. Future studies ought to consider repeated sampling and other intrinsic conditions, including creatinine normalization. More so, research gaps can be highlighted with the following statement: “Prior studies did not investigate chemical compound exposure alongside multiple traffic–related metabolite levels in naïve AMD patients.”

In addition, the studies by Ambati et al. (2013 and Beatty et al. (2000 should be cited. The aforementioned references would contribute insight into key aspects such as the integrative etiopathogenesis of AMD through oxidative stress and consequent damage; thus, buttressing a biological or humanistic perspective on the implication of industrial pollutants in degenerative disease epidemiology.

Figures & Tables
The figures are appropriate, and the legends include the essential components. Several figure annotations possess minor typographical errors, e.g., "Kruskal Wallis" and the symbols for prostaglandins in the Discussion section. The authors must examine the exported text and guarantee consistent typography throughout the panels. Incorporating individual data points as jittered dots on box plots and specifying the precise sample size per group within each panel would enhance interpretability. Additionally, maintaining similar axis scales throughout panels would facilitate comparison. Figures must incorporate superimposed individual data points alongside medians and interquartile ranges to illustrate intra-group variability and to highlight potential outliers.

Tables that summarize fuel composition and ambient volatile organic compound concentrations are beneficial for contextualization. To improve transparency, the authors should include the precise source URLs and retrieval dates for the Tokyo monitoring database and clarify whether the annual means are arithmetic or geometric. The authors should include counts alongside percentages for comorbidities and smoking status in Table 1, and report p-values for group comparisons in a separate column to enable swift evaluation.

Experimental design

Material and Methods
The Methods section clearly entails the study design, analytical methods, protocols, as well as inclusion criteria and exclusion criteria. The study employed a cross-sectional design without randomization; instead, it relied upon successive enrolment to contrast inter-group differences in intrinsic biomarkers of exposure. The authors should endeavor to include essential analytical characteristics for gas chromatography-mass spectrometry and the laboratory quality control plan, including calibration curves, internal standards, limit of detection, and coefficients of variation for each metabolite. The study’s duration lasted around 18 months; hence, cross-sectional study-sampling techniques make a good fit. However, the authors have not declared limitations presented due to diurnal physiologic inconsistencies and the odds of misclassifying exposure risk and route.

The statistical methodology used was an unpaired t-test, while the intergroup comparison employed rank-based correlation. Due to the study’s limited sample size, pairwise comparisons should employ non-parametric methods with appropriate adjustment, and models should account for potential confounders such as age, gender, vocational exposure, tobacco smoking history, diabetes, hypertension, and renal function beyond creatinine normalization. The forward stepwise model has five metabolites and forty instances, which may be marginal for reliable estimates. The study’s criteria for exclusion need to be rid of ambiguity. For example, nuclear cataract scores above grade III were tagged for exclusion; however, the control group included participants with crystalline lens nuclear sclerosis scores above grade III. More elucidation will be necessary to minimize the introduction of bias.

Validity of the findings

Results
The reportedly elevated levels of 2-methyl hippuric acid, mandelic acid, and trans-trans muconic acid among AMD participants in this study represent a novel reportage in ophthalmic health research, and the recognition of mandelic acid as an independent variable in multivariable analysis is significant. The distributional summaries and group comparisons indicate a signal that merits additional examination, and the absence of significant differences among age-related macular degeneration subtypes is clearly documented. The correlation matrix of metabolites is elucidative and indicates common sources or pathways.

The credibility of the findings is corroborated by ambient monitoring data indicating consistent levels of toluene and benzene in the study districts, as well as the established presence of these aromatic chemicals in retail gasoline. However, the study’s academic integrity will be justified by the open-access provision of individual study participants’ data for all urinary metabolites, and by the inclusion of graphs representing data points alongside medians and interquartile ranges for each group. Figures must incorporate superimposed individual data points alongside medians and interquartile ranges to illustrate intra-group variability and to highlight potential outliers.

Discussion
The discussion section comprises of narrative association between exposure risk and elevated findings of 2-methylhippuric acid, mandelic acid, and trans-trans muconic acid. Conventional disease processes, including tissue oxidation, inflammatory pathways via the complement system, and ocular surface irritation, present biologically plausible pathways linking volatile organic compounds to causes of retinal tissue damage.

The Discussion could specifically consider potential confounding factors such as occupation, indoor solvent exposures, proximity to major roadways, household ventilation, as well as active and passive cigarette smoke inhalation. The following statement can be input into the first paragraph of the Limitations subsection. “Confounders such as occupational and domestic factors, proximity to polluted highways, and passive tobacco consumption cannot be exempted; thus, future studies may evaluate these variables and adjust for them appropriately”. Still, sensitivity analysis could be carried out to exclude active tobacco users and properly adjust for age, gender, and systemic risk factors. This may further promote the study’s heterogeneity.
The authors should consider citing studies by Chew et al. (2014) and Ambati et al. (2013) in the third and final paragraphs, respectively. The study by Chew et al (2014) will offer some context on how to mitigate oxidative tissue damage through nutraceutical measures, which may also alter the rate of oxidative damage following organic compound exposure. On the other hand, Ambati et al (2013) offered insights into modern perspectives of AMD pathogenesis via inflammatory pathways involving the complement system. The inclusion of these aforementioned recommended references would further address contextualizing the current lack of support for the connection between exposure and disease etiology, and thus reiterate the significance of preventive approaches.

Conclusion
The study’s conclusions align with its reported findings and do well to correlate research outcomes as a probable causation, hence emphasizing the need for more standardized and repeatable studies. The submission that mandelic acid may be a biomarker is substantiated by the multivariable model, while recognizing that causal inference would only be possible following randomized controlled trial studies. The conclusion section may need the authors’ comments concerning data sharing and reproducibility. In their conclusion, the authors may also suggest subsequent study plans integrating imaging alongside biological markers. E.g,. “Future longitudinal evaluation research on the subject may combine timed urine metabolite assessments with quantitative retinal imaging and free radical measurements for the detailed analysis of exposure-causation correlations.”

Reviewer 2 ·

Basic reporting

In their article, "Metabolites of traffic-related volatile organic compounds in age-related macular degeneration", Tatsuya Mimura and Hidetaka Noma investigate the potential association between exposure to traffic-related volatile organic compounds (VOCs) and the development of age-related macular degeneration (AMD).

The article requires revision, particularly in aligning the presentation of results with the conclusions, as certain discrepancies between the abstract and the main findings may mislead readers.

Main points:
Abstract
Line 37-56 While the Results section provides detailed quantitative evidence, including significant increases in several urinary VOC metabolites among AMD patients, there are certain inconsistencies between the reported findings and the broader conclusions of the article. For example, mandelic acid is highlighted in the abstract as a potential AMD biomarker (OR = 17.20, p < 0.001), but subgroup analysis revealed no significant differences among AMD subtypes after multiple comparison adjustment. Furthermore, the correlations reported for key metabolites (e.g., r = 0.27–0.47) suggest only moderate associations, which may not justify the strength of the causal implications emphasised in the conclusion.

Introduction
The introduction lacks novelty and does not clearly articulate the specific research assumptions relating to the groups under study, particularly given that the reported demographic differences (sex and age) between the groups were statistically insignificant.

Experimental design

The section describing the aims (lines 91–101) should be moved to the beginning of the article, as this provides the essential methodological context needed to interpret the subsequent results.

It should be clarified which specific AMD patients were included in the study, particularly in line 101, where the use of biological markers is mentioned, as the criteria for patient selection remain ambiguous.

Materials and Methods
It is unclear whether the patients were permanent residents of the regions under investigation. Without this clarification, attributing elevated VOC metabolite levels to regional environmental exposure is questionable, and alternative explanations, such as occupational or lifestyle factors, should be considered.

Validity of the findings

Results and Discussion
The paper is rather disorganised, and the statistical analysis is insufficient to support the breadth of conclusions drawn. Notably, the lack of a thorough multivariate analysis incorporating determination coefficients (R²), confidence intervals, and effect size measures hinders the reader's ability to evaluate the contribution of each parameter. These analyses are necessary to clarify which factors independently influence AMD risk, and must be explicitly addressed in both the Results and Discussion sections.

The presentation is weakened by the inclusion of a large number of statistically non-significant findings intermingled with significant results. This dilutes the impact of the statistically robust outcomes and makes the overall interpretation difficult and less convincing.

In the discussion, the authors draw conclusions that are not fully supported by their data. It is unclear whether the measured levels of environmental toxins are significant or how they translate into urinary concentrations, since this information is not provided in the methods section.

The study did not assess any oxidative stress parameters, yet the discussion relies on such mechanisms to interpret the findings. This constitutes an inappropriate extrapolation beyond the presented results and undermines the validity of the conclusions.

Additional comments

Overall, the manuscript requires thorough revision before it can be considered for publication. While the study presents interesting data, significant issues regarding the design of the study, the clarity of the methodology, the statistical analysis, and the interpretation of the results must be addressed. In its current form, the paper is not suitable for publication and should only be reconsidered after major revisions have been made.

---

## Round 0.2 · accepted · Accept

· Academic Editor

Accept

This revised version is suitable for publication in PeerJ.

Reviewer 1 ·

Basic reporting

Title & Abstract
The title of the paper is suitable for this article. The revised abstract has been improved. The authors have addressed the minor issue stated in the previous review.

Introduction
The aim of the study is clear, which is based on the connection between internal exposure to volatile organic chemicals in AMD. The manuscript now mentions the biological causative effect in a more persuasive manner by describing the immunological and oxidative stress pathways. Mandelic acid has been shown as an exposure marker rather than a disease. The results are indicative that regular urban traffic environment exposures can influence ocular risk, which should be taken into account when managing AMD patients.
The modified manuscript shows enhanced clarity and maintains a suitable tone for this environmental health study. The Introduction has been improved with relative information regarding immunological pathways and oxidative stress in AMD. The flow of the text has been improved. Minor typographical errors have been corrected throughout the text.

Figures & Tables
The modified figures, legends and tables are fine and sufficiently address the minor points of improvement included in the review.

Experimental design

Material and Methods
The study mentions the prospective cross-sectional design, ethical approvals, registry entries and consecutive enrollment. The inclusion of a cataract control group without AMD has been clearly described and included in this section and explained in the Results and Discussion sections. Quality control characteristics for gas chromatography mass spectrometry is missing which has been appropriately mentioned as a limitation.

Validity of the findings

Results
The revised version includes multi-group comparisons analysis with Kruskal–Wallis, pairwise assessments with unpaired tests, stepwise logistic regression and Pearson correlations for connection with illness state. Intergroup disparities in common confounders have been shown to not be significant. Multivariable adjustments were limited by sample size. The revised version appears to be more transparent and acknowledged the limitations of the cohort analysis. The Results show group characteristics with clear statements of differences for each metabolite, and display the correlation matrix together with logistic regression results. Mandelic acid has been reported to be independently associated with AMD. The district-level monitoring data and composition of gasoline have been put into tables, thus enhancing transparency.

Discussion
The discussion has been improved. It establishes the biological plausibility in complement-associated inflammatory and oxidative stress pathways. Mandelic acid has been properly shown to be an exposure marker independently correlated with illness status based on the interpretation for cross-sectional data. The Limitations report potential residual confounding from indoor solvents, occupation, proximity to roadways, passive smoke, and ventilation. These modifications appropriate address the previous requests.

Conclusion
The study’s conclusions align with its reported findings. The conclusion has been properly toned down based on the results presented in the paper. The modified version adequately addresses issues raised in the previous review.

Reviewer 2 ·

Basic reporting

The authors have thoroughly revised the manuscript, providing the necessary clarifications and addressing all previous comments.
I have no further comments and can therefore accept the article for publication.

Experimental design

The authors have significantly improved the quality of the manuscript, clarified all issues raised in the initial review, and the current version is satisfactory. I have no further comments, and the paper is ready for acceptance.

Validity of the findings

All of the reviewers' concerns have been properly resolved, and the authors have clarified all previously unclear aspects. I have no further suggestions, and the article may now be accepted for publication.

Additional comments

I have no further suggestions, and the article may now be accepted for publication.